# Efficient Approach for Extracting High-Level B-Spline Features from LIDAR Data for Light-Weight Mapping

**DOI:** 10.3390/s22239168

**Published:** 2022-11-25

**Authors:** Muhammad Usman, Ahmad Ali, Abdullah Tahir, Muhammad Zia Ur Rahman, Abdul Manan Khan

**Affiliations:** 1Department of Mechanical, Mechatronics, and Manufacturing Engineering, University of Engineering & Technology, Faisalabad Campus, Faisalabad 38000, Pakistan; 2Department of Mechanical Engineering, Hanbat National University, Deajeon 34158, Republic of Korea

**Keywords:** range sensing, feature detection, light-weight mapping, B-splines, localization

## Abstract

Light-weight and accurate mapping is made possible by high-level feature extraction from sensor readings. In this paper, the high-level B-spline features from a 2D LIDAR are extracted with a faster method as a solution to the mapping problem, making it possible for the robot to interact with its environment while navigating. The computation time of feature extraction is very crucial when mobile robots perform real-time tasks. In addition to the existing assessment measures of B-spline feature extraction methods, the paper also includes a new benchmark time metric for evaluating how well the extracted features perform. For point-to-point association, the most reliable vertex control points of the spline features generated from the hints of low-level point feature FALKO were chosen. The standard three indoor and one outdoor data sets were used for the experiment. The experimental results based on benchmark performance metrics, specifically computation time, show that the presented approach achieves better results than the state-of-the-art methods for extracting B-spline features. The classification of the methods implemented in the B-spline features detection and the algorithms are also presented in the paper.

## 1. Introduction

The simultaneous localization and mapping of SLAM are already acknowledged as forming a discipline; therefore, the optimization of its specific modules (localization and mapping) is the focus of research these days. The main focus has been moved to the optimization and overall assimilation of these components. Localization tasks can be performed efficiently by using the extraction of distinctive regions or point features from the sensor data. The distinctive regions or points allow the computation of signatures for place recognition, which enables effective robot localization. Only a small group of point-features obtained from a local map was used in point-to-point associations for localization purposes. Three state-of-the-art point feature detectors designed for 2D laser scans are FLIRT [1], FALKO [2], and BID [3]. These low-level features are also used as landmarks for a lightweight map. The map represented using point features ignores the structural information of the surroundings and makes it challenging for robots to interact with that environment. The high-resolution structural information can be attained using occupancy grids [4]. However, to create a dense map using occupancy grids, a large state space is required. The high-level features also enable dense mappings with lesser state space requirements. The extraction of numerous high-level geometric features such as lines [5], polylines [6], circle features [7], and curve segments [8] has also been presented in the literature. Due to a single geometric feature extraction model, the approaches still require a more reliable and robust solution for environments with various shapes.

A comprehensive approach to express the curved and straight features has been presented in vision-based Curve SLAM [9]. A stereo camera image of paths was used to detect the left and right edges, and later, these edges were represented as Bezier curves features. The major drawback of using Bezier curves as features is that the degree of the polynomial is confined to the total number of control points. The degree of the polynomial is always one less than the number of control points. Therefore, the multifaceted shapes are hard to represent using a single Bezier curve. Another limitation of the Bezier curve is that displacing any of the control points changes the curve’s shape completely, which makes it tough to manipulate the control point for curve modifications. A better alternative [10] is the utilization of B-splines in the description of straight and curved-shaped data obtained from a 2D laser scan.

The B-spline representation of a scan is found by the segmentation of the scanned data and then approximating the B-spline curve on the segments [10]. However, the segmentation methodology used in [10] results in the extreme breakdown and depletion of scan data. The curve-fitting method [10] also lacks the appropriate choice with respect to the total number of control points and the derived knot vector. In contrast, a better method for segmentation and the active B-spline curve-fitting approach with corner representations is presented in [3]. Although it results in a highly compact representation of the environment but the approach is computationally very expensive.

In this article, we proposed a novel methodology of B-spline feature extraction. In the existing state-of-the-art approach of B-spline feature extractions, the B-spline-based interest point detection (BID) is used for corner detection from laser scans. In BID, all points are tried as the candidates of the interest points by the active B-spline curve-fitting method using its neighbor points. The application of active B-spline approximation on every data point is a time-consuming practice, which greatly increases the time of the B-spline feature representation of the complete scan. The proposed approach is based on FALKO [2] interest point detection, which is a quicker method for corner detection. It reduces the overall execution time of the B-spline approximation of the complete scan. The performance of the novel approach was compared with the state-of-the-art methodology using the set of benchmark metrics presented in [3]. In the experimentation, the standard data sets of different indoor and outdoor environments are used. We not only suggested names for existing extraction methodologies based on the techniques used for the segmentation and the B-spline approximation but the algorithms of all approaches are also presented in our paper.

### Contributions

The main contributions of this article are as follows:We proposed a novel method for the fast extraction of high-level B-spline features, which can be used for the accurate mapping of the environment suited for both curved and straight-line geometric shapes.The computation time of feature extraction is very crucial for mobile robots when they perform real-time tasks. Therefore, we added a new benchmark time metric in the existing set of metrics used for the evaluation of the B-spline feature extraction approach.The classification of B-spline feature extraction approaches based on the techniques used for segmentation, corner detection, and B-spline approximation is also presented in the paper.Finally, we also presented algorithms for all approaches that can be very effective when implementing the procedures.

The remaining paper is organized as follows: Section 2 provides the literature on low-level and high-level features detection approaches for LIDAR data, and Section 3 provides the problem’s definition and methodology, Section 4 introduces B-Spline theory, Section 5 presents B-spline features extraction approaches, and Section 6 explains the experiments and the results. Finally, the conclusion is provided in Section 7.

## 2. Related Work

The robot’s pose and the map are depicted by the modules of SLAM. Mapping is the process of estimating the map, and localization is referred to as the process of estimating the robot’s pose. The localization of landmarks or features, which can be used to determine the exact location of a robot on a global map, is a problem that localization addresses. Environmental features, both artificial and natural, are shown on landmark maps. Smith et al. [11] came up with the ground-breaking solution of landmark-based EKF-SLAM. Although comprehensively studied [12] and more accurate visual keypoint-based place recognition techniques have their own set of difficulties regarding scalability, viewpoint, and slight variations. The object-based localization frameworks [13,14] provide a compromise between accurate keypoint localization and the capacity to take contextual and semantic information into account.

LiDAR-based Localization techniques can be classified and rely mostly on matching geometries. Numerous point registration techniques, the most well-known of which is iterative closest point (ICP) [15], involve a high pose prior and are, hence, unsuitable for global localization. Even though there are methods for global registration that operate outside of the local context [16,17], they still require keeping at least some of the point cloud data. This can be partially concentrated by simply collecting compact descriptors during map construction and localization. There are various low-level features or keypoints, such as the corner points, which have already been proposed specifically for the planer rangefinders. In [18], Bosse proposed the centroids of the clusters and the points of the large positive curvature as point features for the task of place recognition. Li and Olson in [19] suggested an approach that builds upon the method used in image processing: Kanade–Tomasi is a variant of the Harris corner detector. This method was applied to the laser scan data. The scan data were first converted into an image from which the point features were extracted. Later [20], they improved the point feature detector for LIDAR data, which was also based on the computer vision approach by computing the structure tensor of the surface’s normal. The main concern is the proportional change in computational complexity of the image-processing-based approaches with the change in the size of the converted image from the scanned data. In FLIRT [1], a discrete framework of scale-space theory was used for re-scaling the scanned data; then, the point features were searched on different scales depending on the parameters such as the range, normal, or curvature. However, the single parameter-based detection of the point features may result in unstable points in FLIRT. Kallasi et al. proposed a point feature detector FALKO [2], which used the idea of edge intersection. The first step of the corner detection was the rough guess that was based on the evaluation of the parameters of the triangle generated by two boundary data points in a specific neighborhood and the corner point candidate. Therefore, it is likely to select the unstable candidates or will skip the possible candidates of the point features. In the B-spline-based interest point detector BID [3], the idea of neighborhood selection was taken from [2] but used in a better way by applying active B-spline approximation to the neighbor data points for the detection of point features.

Other than low-level feature extraction techniques, high-level feature extraction approaches such as lines and curve segments are also used for LIDAR data. High-level features have been included in many SLAM algorithms to address the limitations of point-based SLAM. Planes, image moments, line segments, office chairs, tables [21,22,23], objects, and rivers are a few examples of high-level features. The ability to compactly create a map of the environment is a desirable quality of the high-level structure. The outline of a door, the structure of an inside hallway, or other objects one may expect to find in the mapped place are represented by lines [24,25]. Additionally, lines give a sparse picture of the surroundings. For instance, each line is represented by only two points [23]: the start and end points of a line segment. The line features have also been implemented by [23] to map the environment using a vision-based sensor. The start and end points were represented by using line segments. High-level feature detectors presented for 2D laser data contain dynamic mappings based on circles and lines features [26], curve features based on adaptive curvature estimation [8], and weighted line fitting [27]. One of the foremost needed properties of high-level features is that they enable a compact method of mapping the structural details of the environment. A generalized method of representing the curve or line feature using the Bezier curves was explained in CurveSLAM [9] to create an efficient map of the stereo vision sensor’s data. The methodology was used to create the straight and curved pathways as well as the yellow road lanes. A more comprehensive method of high-level features description for the varied and complex environments using B-splines was presented in [10]. However, the approach lacks smart segmentation procedures, choosing an initial number of control points and the parametrization for the B-spline’s feature demonstration. In [3], active B-splines features extraction was implemented with efficient segmentation to reduce the wastage of data points and accurate representation of the line or curved segments. However, it is a computationally very expensive and slow approach.

## 3. Problem Definition and Methodology

In 2D LIDAR data, point features such as corners are taken as landmarks in the environment. These landmarks can also be used to represent a light-weight map of the environment. However, the structural information of the environment is not contained in the landmarks based on point features. Therefore, the robots are unable to interact with the actual environment by using the point-feature-based map. The occupancy grids used for map representations in [28] produced the structural details of the environment shown in Figure 1, but every grid requires dedicated memory to store each laser data point. The higher the resolution, the more space will be required to save the map.

Various high-level features have been used in the literature to represent the map for using laser data points; however, these features are used for dedicated shapes such lines or curves. The B-splines are not only used for straight as well curved shapes to represent the map by fewer control points. However, in the literature, the methodologies used for B-spline representation still lacked an efficient approach.

In B-spline extractions, the two main steps are as follows: One is segmentation, and the second is the B-spline approximation of the segments. An additional step involving corner detection was presented in our previous work [3], which enabled not only light-weight representations of the map but also the corner representation. The detailed methodology [3] is presented in Section 5.3. The corner representation can also be used for B-spline associations. However, this [3] approach is computationally very expensive.

We proposed a faster method for representing laser data using B-splines. FALKO-based corner detection [2] significantly reduces the overall execution time of the B-spline representation and does not affect the other performance metrics of the approach. The three steps includes FALKO-based interest point detection, segmentation, and finally, B-spline curve fitting. It is represented in Figure 2.

The detailed explanation of the approach is presented in a later section of the paper. Firstly, the B-spline theory is presented.

## 4. B-Spline Theory

In this section, some basic concepts of B-spline, spline continuity, and the computation of an active B-spline curve to approximate the data points in a plane are presented.

### 4.1. B-Spline Curve

The B-spline curve is represented as a linear combination of basis functions that are also known as B-spline basis functions. An *l*-dimensional curve S(t) composed of basis functions $B_{i,k}$ of degree k(order=k+1) and control points $P_{i}\epsilon R^{l}\left(i=0,\ldots ,n\right)$ can be expressed as a function of parameter tϵR such as
(1)$$S\left(t\right)=\sum_{i=0}^{n}B_{i,k}\left(t\right)P_{i}$$
where basis functions $B_{i,k}$ are defined by the Cox–de Boor recursion formulas [29,30]
(2)$$B_{i,0}\left(t\right)=\left\{\begin{array}{cc} 1, & \mathrm{if}\;\xi_{i}\leq t\leq \xi_{i+1}\\ 0, & \mathrm{otherwise}. \end{array}\right.$$
and for all k>0
(3)$$B_{i,j}\left(t\right)=\frac{t-\xi_{i}}{\xi_{i+j}-\xi_{i}}B_{i,j-1}\left(t\right)+\frac{\xi_{i+j+1}-t}{\xi_{i+j+1}-\xi_{i+1}}B_{i+1,j-1}\left(t\right)$$
where knot vector $\Xi =\xi_{0},\ldots ,\xi_{n+k}$ is any non-decreasing sequence of real numbers. The multiplicity of the extreme values of the knot vector equal to the curve’s order makes it a clamped B-spline; otherwise, it is an unclamped one.

### 4.2. Continuity

With respect to the knot value, the continuity, which is also known as the degree of smoothness of the B-spline of order *O*, is at its maximal value, that is, $C^{O-2}$ continuity of all derivatives up to (O−2)th [31]. Continuity at a knot multiplicity *m* is reduced to $C^{O-m-1}$. In contrast, coinciding *n* consecutive control points reduce the continuity of the B-spline curve by n−1. For example, two control points of the quadratic (O=3) B-spline coincide to form a hinge (discontinuous first derivative) at the point of coincidence on the curve.

### 4.3. Active B-Spline Curve Fitting

Approximating a noise-contaminated or non-uniformly distributed set of data points $X_{s}\epsilon R^{2},s=1,\ldots ,N$ by using an active B-spline curve can be formulated as a non-linear optimization problem. A data point $X_{s}$ that lies on the B-spline must satisfy
(4)$$X_{s}=B_{0,k}\left(t_{s}\right)P_{0}+\ldots +B_{n,k}\left(t_{s}\right)P_{n}$$

The process of finding $t_{s}$ is known as data parameterization. The main approach of approximation is based on the idea of starting an active B-spline with an appropriately chosen initial curve and converging via iterative optimizations towards targeted data points [32]. The knot vector and the order of the curve are assumed to be fixed throughout the fitting procedure. At first, keeping the parameter values constant, control points $P_{i},i=0,\ldots ,n$ are found such that objective function
(5)$$f=\frac{1}{2}\sum_{s=1}^{N}d^{2}\left(S\left(t\right),X_{s}\right)+\lambda f_{req}$$
is minimized, where $d(S\left(t\right),X_{s})$ is the orthogonal distance from data point $X_{s}$ to the point on the initial curve, S(t). This scheme of minimization is called point distance minimization (PDM) [32]. Secondly, control points produced in the minimization step are kept fixed while the data parameterization method is used to select $t_{s}$ such that $St_{s}$ (also known as footpoint of $X_{s}$) is the nearest point on the updated curve from data point $X_{s}$. Regularization term $f_{reg}$ governs the smoothness by looking at the differential Σ and curvature ρ of curve at every point, whereas positive constant λ represents its weight. Differential operations or stretching of the curve are defined as
(6)$$\Sigma =\int_{}^{}{∥S^{{}^{\prime }}\left(t\right)∥}^{2}\;dt\;$$
and the bending or curvature energy, ρ, of the curve is defined as
(7)$$\rho =\int_{}^{}{∥S^{"}\left(t\right)∥}^{2}\;dt\;$$

In our implementation, only the bending energy has been considered. The approximation error of the curve can be calculated as the root mean squared error $E_{curve}$, which is defined as
(8)$$E_{curve}={\left[\frac{1}{m}\sum_{q=1}^{m}{∥S\left(t_{q}\right)-c_{q}∥}^{2}\right]}^{1/2}$$
where *m* is the total number of data points $c_{q}$ upon which curve approximation is applied.

## 5. B-Spline Feature Extraction

In this section, we present the various types and methods of finding the B-spline features from 2D laser scan data. In a laser scan, the state-of-the-art low-level features or corner detectors FALKO [2] and BID [3] can be used to detect stable and view-point-invariant interest points. The high-level B-spline feature extraction approaches are presented in [3,10]. The B-spline feature, which has at least one associated corner (interest point) or vertex in its representation, is defined as an auxiliary B-spline. In contrast, the B-splines that do not have any associated corner (interest point) or vertex in the representation are called bleak B-splines [3]. The B-spline feature extraction involves two important steps that are a segmentation of the scan data and B-spline curve fitting on the segmented data.

### 5.1. Segmentation

The data points obtained from the laser scan may not be represented using a single B-spline feature. Therefore, the first step in the extraction of high-level B-spline features is the segmentation step. The B-spline feature extraction approaches can be categorized based on the segmentation methodology they adopt.

#### 5.1.1. Relative Position and Orientation-Based Segmentation (RPOS)

The segmentation methodology based on the relative position and orientation between two consecutive data points was presented in [10]. It can be called the relative position and orientation-based segmentation (RPOS). The RPOS is centered on the evaluation of the relative positions of two successive laser data points, as shown in Figure 3, and mathematically represented as follows.
(9)$$p_{i}=d_{i}-d_{i-1}$$

Then, the following assessments were performed.
(10)$$|\alpha_{i}|\leq \alpha_{max}↔\cos\left(\alpha_{max}\right)\leq \cos\left(\alpha_{i}\right)$$
(11)$$max(∥p_{i}∥,∥p_{i+1}∥)\leq \eta min(∥p_{i}∥,∥p_{i+1}∥)$$

When a set of successive data points fulfills both relationships described above, they are taken to be part of the same feature. In RPOS, the proposed value for angular threshold $\alpha_{max}=\left[0,\pi /4\right]$ results in the extraction of bleak segments only. The larger values of $\alpha_{max}$ may associate the corner points in the segment, but it results in an incorrect and excessive number of corners identified in scanned data [10]. Therefore, the values for parameters $\alpha_{max}=\pi /4$ and η=1.5 were chosen for implementation.

#### 5.1.2. Segmentation Using Varying Euclidean Distances (SVEDs)

Another segmentation approach presented in [3] is based on Euclidean cluster extraction [33], as shown in Figure 4. Instead of using a constant distance [33], a varying Euclidean distance threshold was used in [3]. It can be called segmentation using varying Euclidean distance SVED-based cluster extraction. For every point, the neighborhood-radius $r_{q}$ is the function of the datapoint distance $∥d_{q}∥$ from a sensor’s origin, and it is represented in [2] as
(12)$$r_{q}=a\exp(b∥d_{q}∥)\;$$

Parameters a=0.2 and b=0.07 were selected for scan ranges between 0.5 and 30m. Radius $r_{q}$ is taken as the varying euclidean distance. The SVED approach may produce both auxiliary and bleak segments [3], as shown in Figure 5.

### 5.2. B-Spline Features

The flexibility of splines curves to approximate noise-contaminated data is one of their main attractions. Every segment in a scan is represented by the B-spline curve approximation. The two curve-fitting approaches for LIDAR data are presented. The first is a least-squares B-spline approximation LSBA in [10], and tje other is point distance minimization PDM in [3].

#### 5.2.1. Least-Squares B-Spline Approximation (LSBA)

The B-spline curve fitting of a set of data points $X_{s}\epsilon R^{2},s=1,\ldots ,N$ by using a least-squares solution can be devised as an approximation problem. If a data point, $X_{s}$, lies on the B-spline, then it must satisfy Equation (Equation 13) as follows:(13)$$X_{s}=B_{0,k}\left(t_{s}\right)P_{0}+\ldots +B_{n,k}\left(t_{s}\right)P_{n}$$

The problem can be represented by using the equations as follows:(14)$$X=BP\left\{\begin{array}{c} X={\left[\begin{array}{cccc} X_{0} & X_{1} & \ldots & X_{N} \end{array}\right]}^{T}\\ P={\left[\begin{array}{cccc} P_{0} & P_{1} & \ldots & P_{N} \end{array}\right]}^{T}\\ B=\left[\begin{array}{ccc} B_{0,k}\left(t_{0}\right) & \ldots & B_{n,k}\left(t_{0}\right)\\ \ldots & \ddots & \ldots \\ B_{0,k}\left(t_{m}\right) & \ldots & B_{n,k}\left(t_{m}\right) \end{array}\right] \end{array}\right.$$
where matrix *B* is the collocation matrix. The process of finding $t_{s}$, which is the position of datapoint along the curve, is known as data parameterization. The cumulated chord length can be used to find the position between consecutive datapoints as follows.
(15)$$\left\{\begin{array}{c} t_{0}=0\\ t_{j}=\sum_{s=1}^{i}∥X_{s}-X_{s-1}∥ \end{array}\right.$$

The total length of the curve, *l*, which is the maximum value of the knot vector, can be calculated as follows.
(16)$$l=\sum_{s=1}^{m}∥X_{s}-X_{s-1}∥$$

Finally, the least-squares solution of the approximation problem can be calculated by using the pseudoinverse matrix of *B*.
(17)$$P={\left[B^{T}B\right]}^{-1}B^{T}X$$

In the case of bleak segments, the order (or degree) of the B-spline curve, the number of control points, and the parameter values along the curve are predefined to approximate the segments using the least-squares solution [10]. It provides a solution for bleak segments only, because in this approximation, the corner position cannot be provided, and only the number of control points is given.

#### 5.2.2. Point Distance Minimization (PDM)

Another method for representing the segments using B-splines is active B-spline curve fitting, which is presented in Section 4. The approach involves two steps. The first one is the initialization of a B-spline curve of a specific order and shape using a known position and the number of control points and secondly, the point distance minimization (PDM) scheme [32] is applied to the initial curve to approximate the segments. The main appeal of this approach is that not only bleak segments but the point distance minimization scheme can also be used to approximate auxiliary segments that have at least one associated corner detected in the laser scan, as shown in Figure 5.

The significant idea of the auxiliary segments is the representation of the corners. In the quadratic B-spline curve, a corner can be represented by the knot multiplicity $M_{knot}=2$ or the control point multiplicity $M_{cp}=2$ [31]. The knot multiplicity approach was used in [3] to minimize the number of control points used in the B-spline’s representation. However, representing the corner using control point multiplicity $M_{cp}$ results in the part of the curve on either side of the corner constrained to be linear [31]. The importance of linearity constraints on either sides of the corner is significant for the representation of corners in most of the building’s structures.

The control points in the auxiliary segment are categorized into three types. The control point used in the representation of the corner is defined as a vertex control point [3]. The boundary control points are initialized at the boundary data points, whereas the precision control points are used in the representation of the fragments [3]. The fragment is the region of the B-spline curve between two vertex control points or the region between the boundary control point and the vertex control point [3]. The precision control points define the accuracy of the B-spline approximation of a segment. Therefore, the number of precision control points was proposed to be chosen as a function of the Euclidean distance between the fragment’s endpoints [3]. Once the curve is initialized using the control points, the PDM scheme is applied to approximate the segment.

### 5.3. Extraction Methodologies

Various approaches can be used in the extraction of B-spline features from 2D laser scan data. The approaches are categorized on the basis of the phenomena they adopt.

#### 5.3.1. RPOS-LSBA

The RPOS-LSBA [10] used a typical approach involving B-spline extraction, which involved the segmentation of the laser scan and then B-spline curve fitting on the segments. In the segmentation step, it employed the relative position and orientation-based segmentation RPOS, which is based on the relative position of $d_{i}$ and orientation $\alpha_{i}$ of the two consecutive scan data points, as shown in Figure 3. As segmentation relies on the geometrical (angle and distance) property of the two very next neighboring data points, therefore, a distant point from the neighbors not only creates a new segment but also limits the upcoming neighbor in becoming a part of a segment rather than isolating it. This increases wasted data points. Segments containing less than five data points are considered wasted. The second step is the application of the least-square solution [10] to the segments obtained in RPOS to obtain a B-spline approximation. It is named least square B-spline approximation LSBA. This B-spline feature extraction approach not only increases wasted data points but also lacks a representation of stable points (corners) in the scan. The B-spline feature extraction based on RPOS-LSBA is presented in Algorithm 1.
**Algorithm 1:** RPOS-LSBA Features EtractionSEGMENTATIONrpos_seg$\left(r_{th},\alpha_{max}\right)$ // position and orientation thresholds**for** each datapoint $d_{i}$ in scan S **do**    **if** position_and_orientation $<(r_{th},\alpha_{max})$
**then**        $\phi_{n}\leftarrow d_{i}$    **end if****end for****return**ϕ // creates and return Bleak segmentsB-SPLINE APPROXIMATIONlsba_curve(ϕ)**for** each segment $\phi_{n}$ in scan S **do**    BSplines= least_square_Approx$\left(P_{c},Order_{curve},\phi_{n}\right)$ // no. of control points $P_{c}$**end for****return**BSplines

#### 5.3.2. BID-SVED-PDM

Another method for B-spline feature extraction was presented in [3]. In this approach, an additional step of corner detection was proposed. Firstly, the stable and state-of-the-art B-spline-based interest point detector BID [3] was used for corner detection, as shown in Figure 6. Roughly, a set of a uniform number of neighbor data points $N(c_{q})$ is acquired using radius $r_{q}=a\exp(b∥d_{q}∥)$. The clamped and uniform quadratic B-spline curve is initialized with four control points evenly spaced along the axis defined by the largest eigenvector of the covariance matrix of $N(c_{q})$, as shown in red color in Figure 6 (left). Then, the point distance minimization (PDM) method is used to estimate the shape specified by the data points, as in Figure 6 (middle). Finally, the inverse of the normalized Euclidean distance, $N_{q}$, among two median control points, $P_{q1}$ and $P_{q2}$, of the estimated curve, $S_{q}\left(t\right)$, is taken as the measure of corner occurrence as in Figure 6 (right).
(18)$$N_{q}=\frac{∥P_{q1}-P_{q2}∥}{\sum_{i=0}^{2}∥P_{qi}-P_{q(i+1)}∥}$$

The local maxima of $N_{q}^{-1}$ beyond threshold $T_{th}$ can be employed to detect the indices of scan points and ultimately corner point $I_{p}$ by using non-maxima suppression NMS. The data points with $N_{q}$ values 0.249 and 0.250 in Figure 6 (right) are the detected interest points.

In the second step, segmentation using a varying Euclidean distance-based cluster extraction (SVED) approach was used [3] and derived from [33], which produces segments with a minimum wastage of data points. Finally, the active B-spline curve approximation based on point distance minimization PDM [32] was used for the representation of the segments using B-spline features. The active B-spline approach enables the representation of detected corners in the laser scan. Algorithm 2 shows the B-spline feature extraction based on BID-SVED-PDM.
**Algorithm 2:** BID-SVED-PDM Feature ExtractionCORNER DETECTIONbid_$I_{p}$$\left(T_{th}\right)$ // $T_{th}$ normalized Euclidean distance threshold [3]**for** each datapoint $d_{i}$ with its neighbors in scan S **do**    **if** $E_{q}^{-1}>T_{th}$
**then**        $I_{p}\leftarrow d_{q}$    **end if****end for**$I\leftarrow Non\_Maxima\_Suppression(I_{p})$**return***I* // returns I set of interest pointsSEGMENTATIONsved_seg$\left(r_{q}\right)$ // varying Euclidean radius $r_{q}$**for** each datapoint $d_{i}$ in scan S **do**    **if** $Euclidean\_distance\leq r_{q}$
**then**        $\phi_{n}\leftarrow d_{q}$    **end if****end for****return**ϕ // creates and return Bleak segmentsACTIVE B-SPLINE APPROXIMATIONactive_Bspline_curve(ϕ)**for** each segment $\phi_{n}$ in scan S **do**    (Aux,Bleak)= find_aux_bleak$\left(\phi_{i},I_{p}\right)$ // search for Auxiliary and Bleak Segments**end for**Bleak Segments B-spline Approximation**for** each bleak segment $\phi_{n}$ in scan S **do**    $B_{cp}$ ← Find_Boundary_Control_Points $\left(\phi_{n}\right)$    $P_{cp}$ ← Find_Precision_Control_Points $\left(B_{cp}\right)$    $curve_{init}$ ← Curve_Initialization $\left(B_{cp},P_{cp},order_{curve}\right)$    Bspline ← point_distance_minimization $\left(curve_{init},\phi_{i}\right)$**end for**Auxiliary Segments B-spline Approximation**for** each Auxiliary segment $\phi_{n}$ in scan S **do**    $B_{cp},V_{cp}$ ← Find_Boundary_Vertex_Cntrl_Pnts $\left(\phi_{n},I_{p}\right)$    $P_{cp}$ ← Find_Precision_Control_Points $\left(B_{cp}\right)$    $curve_{init}$ ← Curve_Initialization $\left(B_{cp},V_{cp},P_{cp},order_{curve}\right)$    Bspline ← point_distance_minimization $\left(curve_{init},\phi_{i}\right)$**end for****return**BSplines

#### 5.3.3. FALKO-SVED-PDM

The B-spline-based interest point or corner detection is a computationally very expensive and time-consuming approach. Therefore, in this section, we proposed to use the fast adaptive laser keypoint orientation-invariant FALKO [2] for corner detection. It manipulates the simple idea of edge crossing in 2D range data, as shown in Figure 7, which makes it faster. After finding the potential corner candidates based on the minimum number of neighbors (cardinality) on each side, they are evaluated geometrically (by measuring the height of the triangle as in Figure 7) to obtain a rough approximation of the corner. Then, for each candidate point, a cornerness score is computed.
(19)$$score\left(p_{i}\right)=score_{L}\left(p_{i}\right)+score_{R}\left(p_{i}\right)$$

This score function measures the alignment of the two point sets on each side of the candidate point as follows:(20)$$score_{L}\left(p_{i}\right)=\sum_{h=i-1}^{j_{min}}\sum_{k=h-1}^{j_{min}}\left|d_{\theta }\left(\phi_{h}\right),\phi_{k}\right|$$
(21)$$score_{L}\left(p_{i}\right)=\sum_{h=i+1}^{j_{min}}\sum_{k=h+1}^{j_{min}}\left|d_{\theta }\left(\phi_{h}\right),\phi_{k}\right|$$
where $d_{\theta }$ is the distance function of the quantized orientations $\phi_{1}$ and $\phi_{2}$ concerning the candidate point as
(22)$$d_{\theta }\left(\phi_{1},\phi_{2}\right)=\left(\phi_{1}+\phi_{2}+\frac{s_{n}}{2}\right)mod\left(s_{n}\right)-\frac{s_{n}}{2}$$
where the number of circular sectors in the polar grid is represented as $s_{n}$. Secondly, segmentation SVED and, finally, the active B-spline curve approximation using PDM are employed. This approach can be named FALKO-SVED-PDM. Although, FALKO detects a few suporious interest points, it is fast; therefore, the proposed methodology considerably reduces the overall execution time of the extraction of the B-spline curve. The FALKO-based corner detection algorithm is shown in Algorithm 3.
**Algorithm 3:** FALKO-Based Corner Detection.CORNER DETECTIONfalko_$I_{p}$$\left(score_{th}\right)$ // $score_{th}$ FALKO score threshold**for** each datapoint $d_{i}$ using its neighbors in scan S **do**    **if**
cardinality≥2
**then**        potential_candidate ← $d_{q}$    **end if****end for****for** each potential candidate $d_{c}$ with its neighbors in scan S **do**    compute_corner_score()    **if** $corner\_score\geq score_{th}$
**then**        $I_{p}\leftarrow d_{q}$    **end if**    $I\leftarrow Non\_Maxima\_Suppression(I_{p})$**    return***I* // returns I set of interest points**end for**

## 6. Experiments

In this section, the experimental setup and results are presented based on the performance evaluation of the B-spline feature extraction approach for compact and faster mapping using B-spline features extracted from the 2D range data. The data sets, three indoor (Intel, Fr079, and MIT-csail) and one outdoor (Fr-clinic), provided by [1] are tested in our experiments. The data sets given in [1] contain both the corrected ground truth and the corresponding original scans.

### 6.1. Experimental Setup

The method of B-spline features extraction is evaluated based on the four metrics of performance, which are retrievability, compactness, accuracy, and time. Although the first three metrics were proposed in [3], the fourth metric time is proposed in this paper as the execution time becomes very important when the robot performs real time tasks.

#### 6.1.1. Retrievability (Γ)

It determines the extent of scan data points that are allotted to the extracted B-spline features. It is described by the percentage of the data points designated for the B-spline features over the total number of data points N in a scan. It can be represented as
(23)$$\Gamma =\frac{1}{N}\sum_{i=1}^{m}\Xi_{\Phi_{i}}$$
where Ξ shows the number of data points in a non-discarded segment (which has more than five data points), and *m* represents the total number of non-discarded segments in a scan. It can be used as the measure of the wasted data points in a scan.

#### 6.1.2. Compactness (η)

The second metric of performance is the compactness η of the depiction of a laser scan using B-spline features. It is calculated by the ratio of the total number of control points of B-splines to the total number of data points allocated to those B-splines in a scan as follows:(24)$$\eta =\frac{\sum_{i=1}^{m}N_{cp_{i}}}{\sum_{i=1}^{m}\Xi_{\Phi_{i}}}$$
where $N_{cp_{i}}$ shows the number of control points in the ith B-spline feature of $\phi_{i}$ in a scan. The smaller value of η means that it is a more precise and compact representation of a scan. This means that lesser control points are used in the representation of a scan.

#### 6.1.3. Accuracy (∧)

The accuracy ∧ of the representation of the scan using B-spline features is assessed by the fitting error, $E_{curve}$, of the curves. The approximation error, $E_{curve}$, of the extracted features in the scan is calculated and then the average of the approximation error was taken as a measure of accuracy. It can be represented as follows
(25)$$\wedge =\frac{1}{m}\sum_{i=1}^{m}E_{curve_{i}}$$
where *m* shows the number of B-spline features of non-discarded segments, and $E_{curve_{i}}$ is the approximation error of the ith feature in the scan.

#### 6.1.4. Execution Time (t)

The execution time *t* of the B-spline curve approximation is the time taken by the method in the representation of a laser scan using B-spline features. The B-spline extraction time of complete data set is averaged over the total number of scan in the set to determine the average time of one scan.

### 6.2. Results Evaluation

By contrasting it with the approaches described in [2,10] using the performance criteria presented, the suggested B-spline features extraction approach was evaluated. Additionally, feature extraction execution time criteria were also proposed to assess the performance of all approaches. For segmentation, the parameter values were set exactly as suggested in [10], but for feature representation, a least-square uniform quadratic B-spline approximation was employed. In the SVED approach of segmentation, the same values of the parameters were set for varying radius $r_{q}$, as shown in [3]. The same parameter values that were chosen for FALKO [2] and BID [3] were set for the assessment of both methodologies. Only the Bleak segments are produced by the RPOS method described in [10]. Therefore, four control points were used in the approximation of all B-splines in RPOS-LSBA-based approach.

Figure 8 and Figure 9 show the segmentation results of RPOS and SVED approaches, respectivey. The noisy scan was taken from the Intel data set. For the same scan, RPOS produces a lot more wasted data points (red markers), as shown in Figure 6, whereas SVED produce few wasted data points (red markers), as shown in Figure 9. The green and purple markers are shown to represent two separate and consecutive segments. The wasted data points will not be used in the representation of B-splines; therefore, it will be taken as empty space in the environment.

Figure 10a shows the implementation of RPOS-LSBA methodology representing the retrievability and compactness performance measures. In a noise-free scan taken as a sample, 34 out of 180 data points were wasted, whereas in SVED, only 14 data points were wasted, as shown in Figure 10b,c. The number of wasted data points increases significantly in the case of noise scan data, as shown in Figure 6. The retrievability increased significantly in BID-SVED-PDM and FALKO-SVED-PDM. Figure 10a also shows that 40 control points were used in the representation of 144 data points, whereas 43 control points were used in the representation of 166 data points, as shown in Figure 10b. In noisy scan data, a significantly larger value of η is achieved while using RPOS-LSBA, which shows poor performance.

Figure 11 shows the resultant map of the Intel data set created using the FALKO-SVED-PDM approach. It not only accuratly approximated the straight but also the curved features. The map contains the B-spline segments with approximation error $E_{curve}\leq 0.01$.

For all data sets, the graphical results of all performance measures are shown in Figure 9 every time the datasets were scanned and all four metrics were computed. After that, the outcomes are averaged across all scans.

In all four data sets, the state-of-the-art [3] approach results in two outstanding metrics: retrievability and compactness; however [3], they have very poor execution times. The bar graph demonstrates that our proposed approach’s performance significantly improved in terms of execution times when compared with the state-of-the-art approach [3], as shown in Figure 12d. In terms of retrievability and compactness, the proposed approach’s performance is comparable with the state-of-the-art method and significantly better than the RPOS-LSBA. However, the performance of FALKO-SVED-PDM and BID-SVED-PDM in the approximation error (accuracy) is slightly low but within an acceptable range.

Table 1 lists the summarized outcomes of the three methods. The results reveal our proposed approach’s superior performance in contrast to the RPOS-LSBA [10] in terms of retrievability and compactness across all datasets, which is analogous to BID-SVED-PDM [3]. However, in the three data sets, the computation time of the proposed method has been reduced by more than 300% if we compare it with the execution time of the state-of-the-art approach [3]. Although the proposed method’s approximation accuracy was slightly low, the overall curve-fitting results are still good and are within the acceptable range. All tests were performed on an Intel Core i7-750 CPU at 2.70 GHz 8 GB RAM with an Ubuntu 16.04 LTS and ROS Kinetic operating system.

## 7. Conclusions

The properties of high-level features and low-level point features, respectively, were used in this article to address the mapping and localization issues for the 2D range data. We presented the classification of the B-spline feature extraction approaches based on their techniques. We proposed a method for the fast and accurate mapping of the environment based on high-level characteristics such as B-splines suited for both curved and straight-line geometric shapes. We also proposed a benchmark time metric for the evaluation of the proposed methodology and compared it with the state-of-the-art methods. It was found that the performance of the proposed approach was comparable in retrievability and compactness for all indoor and outdoor data sets compared with the state-of-the-art approach [3] of B-spline feature extractions; however, this was obtained at the cost of a slight reduction in the accuracy of the curves that was still satisfactory. The time taken for the approximation of B-spline features was greatly reduced in the proposed method compared with the state-of-the-art methodology. We finally presented algorithms for all methods that were very effective when implementing the procedure. In future, we will explore more efficient methods for B-spline approximations, test our method for B-spline associations, and expand the application of our work for the 3D range sensors as well as for the visual SLAM.

## Figures and Tables

**Figure 1 sensors-22-09168-f001:**
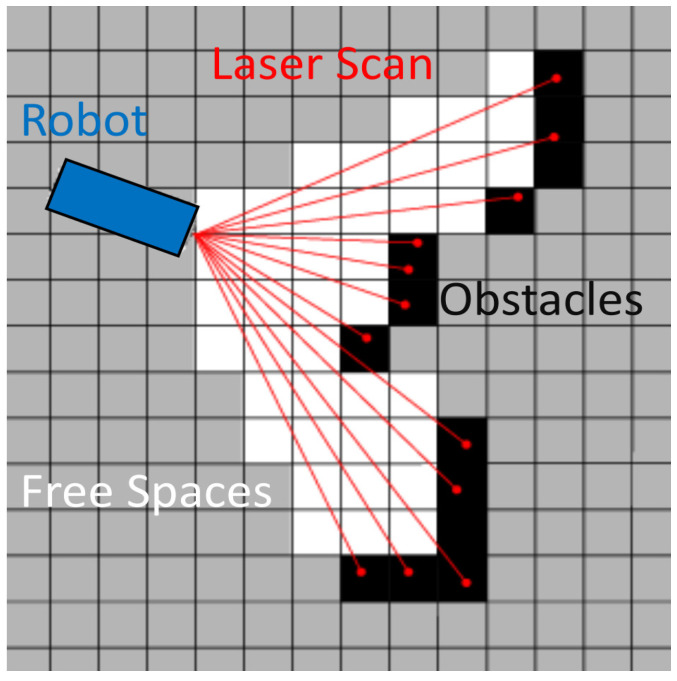
The occupancy grids used in [28] for a laser scan.

**Figure 2 sensors-22-09168-f002:**
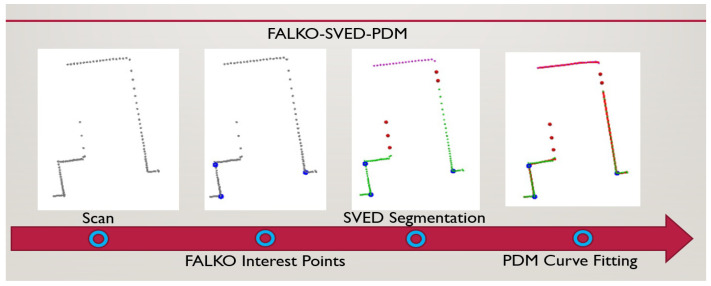
The three steps of proposed approach applied on a scan: FALKO interest point detection, segmentation, and finally, B-spline curve fitting. Note: Control points of B-splines are not shown in the figure.

**Figure 3 sensors-22-09168-f003:**
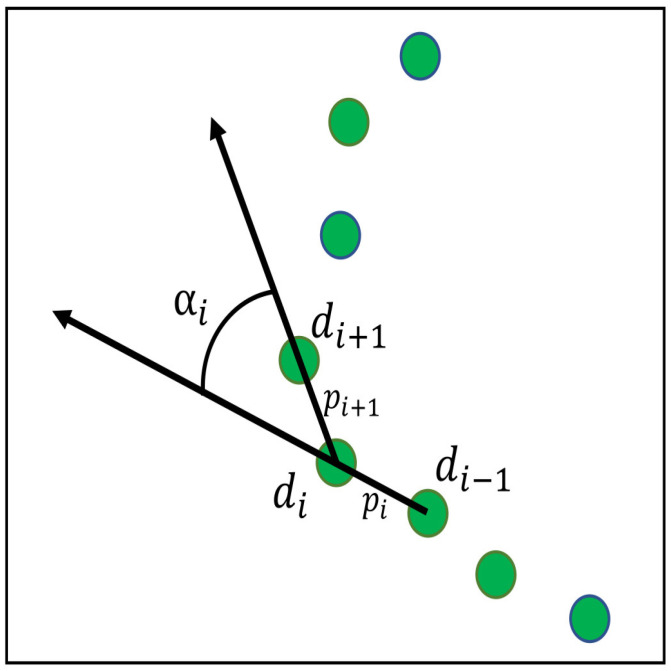
An RPOS-based segmentation presented in [10].

**Figure 4 sensors-22-09168-f004:**
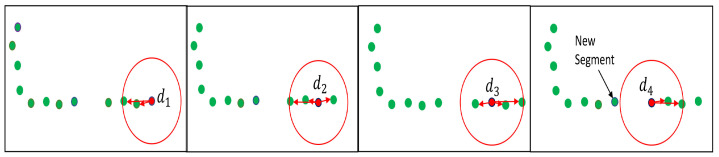
The Euclidean cluster extraction methodology [33] adopted in [3] with varying $r_{q}$.

**Figure 5 sensors-22-09168-f005:**
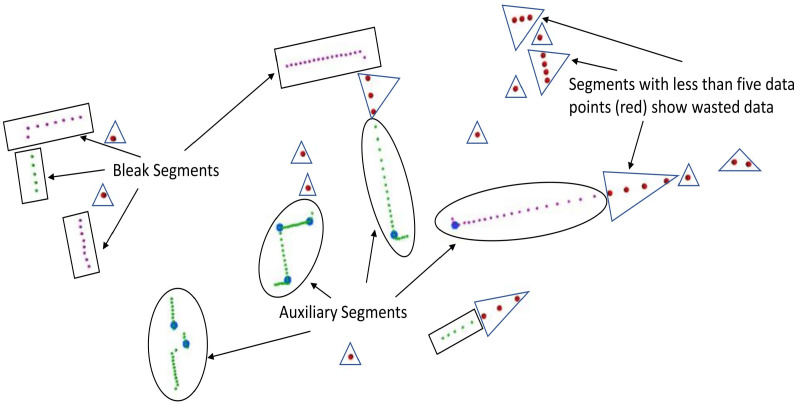
An SVED segmentation-based approach containing the Bleak segment (purple datapoints with no interest points in blue) and auxiliary segments (with associated Interest points in blue color) where the wasted data points are shown in red. The BID-based interest point detection approach was used to detect corners (shown in blue).

**Figure 6 sensors-22-09168-f006:**
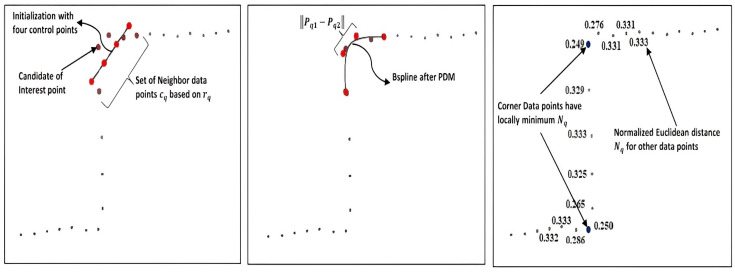
The B-spline-based interest point detection BID method with initialization step (**left**), PDM (**middle**) step for the corner data point, and the normalized Euclidean distance $N_{q}$ values **(right)** calculated for candidates of the interest points around the actual corner points [3]. The four control points are represented in red color.

**Figure 7 sensors-22-09168-f007:**
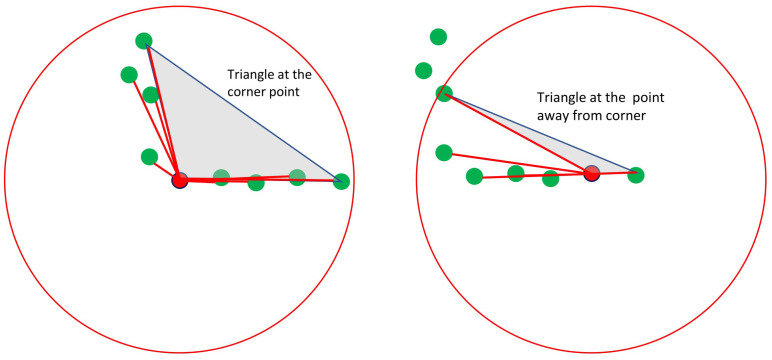
A FALKO-based corner detection example presented in [2].

**Figure 8 sensors-22-09168-f008:**
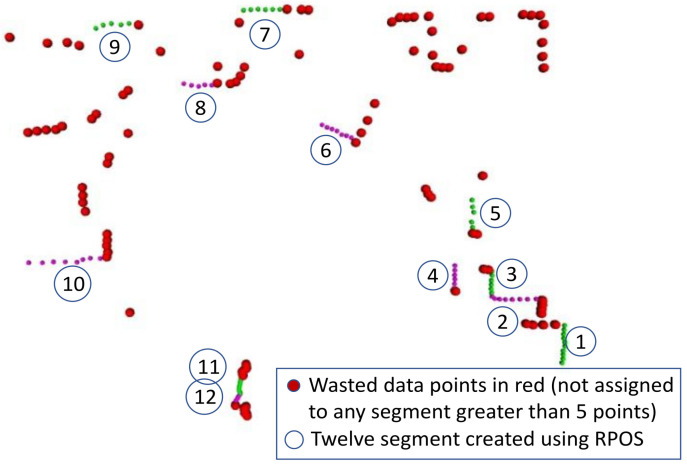
The relative position and orientation-based segmentation RPOS approach with red marks representing wasted datapoints, which are not part of any segment in the scan. A total of 12 green and purple segments were created by this approach. The green and purple colors are used to show the data points of two consecutive segments.

**Figure 9 sensors-22-09168-f009:**
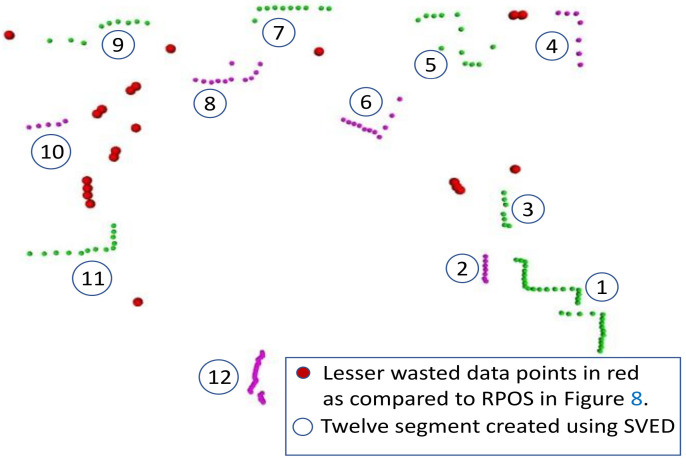
The segmentation using varying Euclidean-distance-based cluster extraction (SVED) approach with red marks representing lesser wasted datapoints, which are not part of any segment in the scan. The green and purple colors are used to show the data points of two consecutive segments.

**Figure 10 sensors-22-09168-f010:**
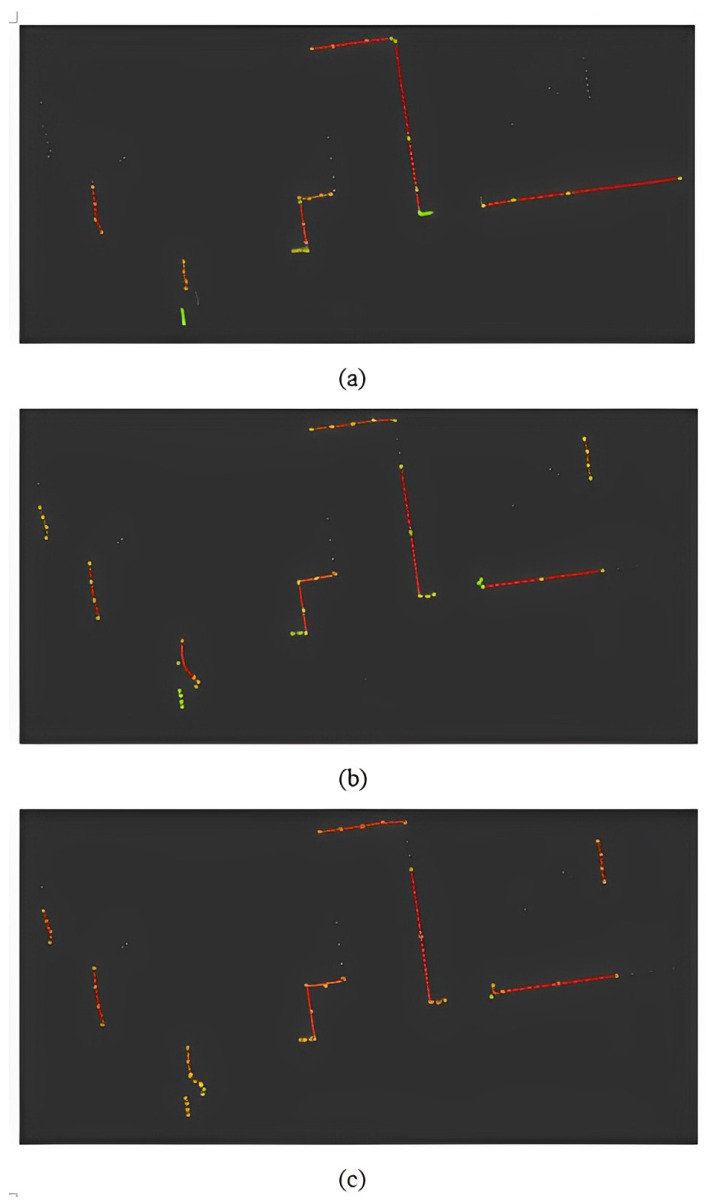
The retrievability and compactness results of (**a**) RPOS-LSBA, (**b**) BID-SVED-PDM, and (**c**) FALKO-SVED-PDM feature extraction approaches. White datapoints show wasted data, and green points are the control points used to represent the B-splines in the scan.

**Figure 11 sensors-22-09168-f011:**
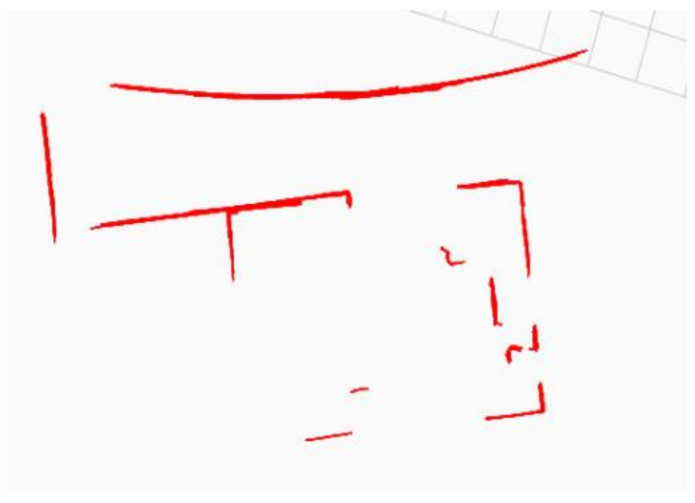
The map of a part of the Intel data set showing the curved and straight walls represented using FALKO-SVED-PDM. The map contains the B-spline segments with approximation error $E_{curve}\leq 0.01$.

**Figure 12 sensors-22-09168-f012:**
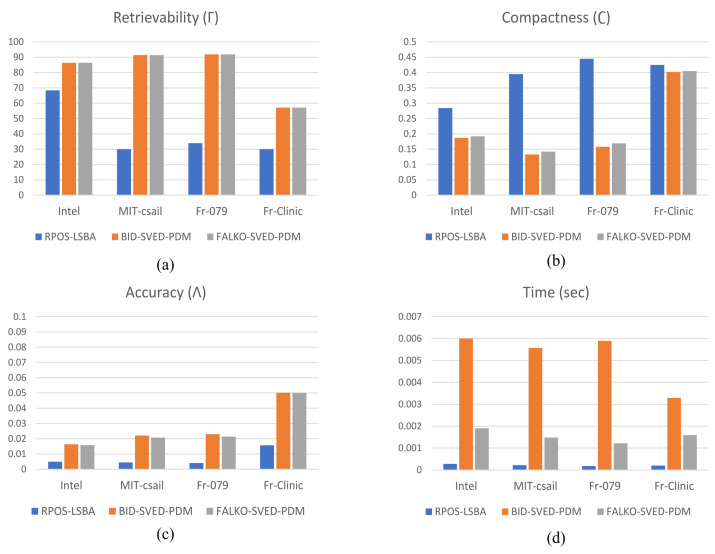
The retrievability (**a**), compactness (**b**), accuracy (**c**), and execution time (**d**) results of complete data sets for all three approaches using all three methods. The vertical axis shows the quantitative measure of the performance of all three methods, and the horizontal axis shows data set types.

**Table 1 sensors-22-09168-t001:** B-spline features assessement.

Data Set	Methodology	Γ	η	∧	*t* (s)
Intel	RPOS-LSBA BID-SVED-PDM FALKO-SVED-PDM	68.4 86.4 86.4	0.284 0.187 0.192	0.0049 0.0163 0.0158	0.00028 0.00600 0.00190
MIT-csail	RPOS-LSBA BID-SVED-PDM FALKO-SVED-PDM	30.0 91.4 91.4	0.395 0.133 0.142	0.0045 0.0221 0.0207	0.00022 0.00557 0.00148
Fr-079	RPOS-LSBA BID-SVED-PDM FALKO-SVED-PDM	33.9 91.9 91.9	0.445 0.158 0.169	0.0041 0.0230 0.0214	0.00018 0.00590 0.00122
Fr-Clinic (1000 Scan)	RPOS-LSBA BID-SVED-PDM FALKO-SVED-PDM	30.0 57.1 57.1	0.425 0.402 0.405	0.0157 0.0501 0.0501	0.00020 0.00329 0.00159

## Data Availability

Anlayzed datasets in this paper were also used in [1,2,3]. The datasets with corrected poses were taken from https://radish.sourceforge.net/index.php (accessed on 3 February 2019).

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
