# Peer review of "Efficient Approach for Extracting High-Level B-Spline Features from LIDAR Data for Light-Weight Mapping"

_sensors, 2022, doi:10.3390/s22239168_

Round 1
Reviewer 1 Report
The paper is generally well-written. On the other hand, the paper should be improved by considering the following concerns:
+The main contributions of the paper should be given clearly as a subsection in the introduction section.
+ Before the Section of "B-spline Theory" where a background is given, the manuscript should present a section of system model and problem definition as Section 3. Then, the background and the rest of paper should be given.
+Figure 4 is not explanatory. What should be understood from the subfigures of Figure 4?
+Similarly, Figure 6 and 7 are not explanatory.
+The subfigures in Figure 9 should be drawn via professional tools.
Author Response
We are very grateful to the Editors and Reviewers for their comments. We have followed them and included our itemized responses below in red following the comment. Our corresponding changes in the manuscript in response to the reviewer’s comments are also highlighted in red.
Thank you for your positive comments. We tried to address all your concerns and done our best effort to strengthen the problem statement and motivation to be more clearly highlighted.

Reviewer 2 Report
1. Authors aim to propose a light-weigth and accurate mapping approach with 2D Lidar data. However, the actually present a line estimator based on B-splines, which does not matches with what they claim they propose.
2. It should be compared against well-ack methods for 2D Lidar mapping as I suggested: i.e. Gmapping as a minimum, from G. Grissetti et al.
3. The present a supposedly more efficient line estimator vs a previous work cited as [3]. However, only one contribution is added with respect to that work.
4. As already commented, they should be compare their approach, in general efficiency and accuracy, against other well-ack lidar-based mapping approaches such as Gmapping, Li-SAM, GT-SAM, or similar. That would put a valid proof that the approach is comparable in terms of efficiency to the state-of-the-art.
5. Other publicly available datasets could also be tested.
Author Response

(The authors gave the same response as above.)

Round 2
Reviewer 1 Report
My concerns on the previous version of the manuscript have been addressed sufficiently. Now, the paper paper is acceptable for possible publication.
Author Response
Dear Reviewer,
We are very grateful to you for accepting our paper. Thank you for your time.

Reviewer 2 Report
The approach is now clearer focused on the line descriptor. However, for the near future would be highly needed to compare with similar approaches as pointed in comment no.5.
Please revise resolution/quality of figures, ie. Fig.12, 9 & 8 (unreadable legends)
Author Response
Dear Reviewer,
We are very grateful to you for the positive comments. Thank you for your time.
